# Application of Endoxylanases of *Bacillus halodurans* for Producing Xylooligosaccharides from Empty Fruit Bunch

**Chanakan Thirametoakkhara** [1], **Yi-Cheng Hong** [1], **Nuttapol Lerkkasemsan** [2], **Jian-Mao Shih** [1], **Chien-Yen Chen** [3] and **Wen-Chien Lee** [1,*]

1 Department of Chemical Engineering, National Chung Cheng University, 168 University Road, Min-Hsiung, Chiayi 62102, Taiwan

2 Department of Chemical Engineering, School of Engineering, King Mongkut's Institute of Technology Ladkrabang, Ladkrabang, Bangkok 10520, Thailand

3 Department of Earth and Environmental Sciences, National Chung Cheng University, 168 University Road, Min-Hsiung, Chiayi 62102, Taiwan

* Correspondence: chmwcl@ccu.edu.tw

**Abstract:** Endo-1,4-β-xylanase catalyzes the random hydrolysis of β-1,4-D-xylosidic bonds in xylan, resulting in the formation of oligomers of xylose. This study aims to demonstrate the promise of endoxylanases from alkaliphilic *Bacillus halodurans* for the production of xylooligosaccharides (XOS) from oil palm empty fruit bunch (EFB) at high pH. Two enzyme preparations were employed: recombinant endoxylanase Xyn45 (GH10 xylanase) and nonrecombinant endoxylanases, a mixture of two extracellular endo-1,4-β-xylanases Xyn45 and Xyn23 (GH11 xylanase) produced by *B. halodurans*. EFB was first treated with an alkaline solution. Then, the dissolved xylan-containing fraction was retained, and a prepared enzyme was added to react at pH 8 to convert xylan into XOS. Compared with the use of only Xyn45, the combined use of Xyn45 and Xyn23 resulted in a higher yield of XOS, suggesting the synergistic effect of the two endoxylanases. The yield of XOS obtained from EFB was as high as 46.77% $\pm$ 1.64% (*w/w*), with the xylobiose-to-xylotriose ratio being 6:5. However, when the enzyme activity dose was low, the product contained more xylotriose than xylobiose. Four probiotic lactobacilli and bifidobacteria grew well on a medium containing XOS from EFB. The presence of XOS increased cell mass and reduced pH, suggesting that XOS promoted the growth of probiotics.

**Keywords:** endo-1,4-β-xylanases; xylooligosaccharides (XOS); empty fruit bunch (EFB); *Bacillus halodurans*; xylan

## 1. Introduction

The alkaliphilic *Bacillus halodurans* can secrete enzymes to break down insoluble xylan, the main component of hemicellulose, into a carbon source for cell growth. According to the whole-genome sequencing data of *B. halodurans* C-125 [1], this alkaliphilic bacterium can synthesize endo-β-1,4-xylanase, acetylxylan esterase, α-L-arabinofuranosidase, α-glucuronidase, β-xylosidase, xylan β-1,4-xylosidase, and a reducing-end xylose-releasing exo-oligoxylanase to completely utilize xylan. During its growth on xylan-containing substrates, *B. halodurans* induces the production of some xylanolytic enzymes and their subsequent secretion into the environment; however, some enzymes are bound by cells [2,3]. Among xylanolytic enzymes, two isozymes of endo-β-1,4-xylanase are considered extracellular because they contain signal peptides. Our previous study indicated that *B. halodurans* BCRC 910501 can synthesize extracellular endo-1,4-β-D-xylanases, which are highly active and could effectively hydrolyze xylan extracted from various types of agricultural waste to produce xylo-oligosaccharides (XOS). These extracellular enzymes have potential for industrial application and are suitable for use in free and immobilized forms [4].

Endo-1,4-β-xylanase (EC 3.2.1.8) is an enzyme belonging to the class of glycoside hydrolases and catalyzes the random hydrolysis of β-1,4-D-xylosidic bonds within xylan [5].

Two endo-1,4-β-xylanases purified from *B. halodurans* BCRC 910501 had molecular weights of 45 and 23 kDa, respectively, and at 37 °C, they exhibited enzymatic activity over the pH range 5.0–11.0 [6]. This bacterium was previously named *Bacillus firmus* [6,7], but later reidentified as *B. halodurans* on the basis of the findings of 16S rDNA gene sequencing. The two cellulase-free endoxylanases are abbreviated Xyn45 and Xyn23, respectively, due to their molecular weights. Xyn45 belongs to glycoside hydrolase family 10 (GH10), which is characterized by high molecular weight and low pI. Xyn23 belongs to family 11 (GH11), which is characterized by low molecular weight and high pI [7]. These enzymes cause the cleavage of glycosidic bonds inside the xylan backbone, reducing the degree of polymerization of substrates and releasing xylose, with the main product being XOS [8–10]. The breakdown of xylan by endoxylanases mainly leads to the production of xylobiose and xylotriose [9]. Because of its relatively large molecular size, the endoxylanase of GH10 is more capable of attacking the nonreducing end of the glycosidic bond than that of GH11. The amino acid sequences of Xyn45 and Xyn23 are identical to those encoded by two endoxylanase genes in *B. halodurans* C-125: BH2120 and BH0899, respectively [11]. The gene (*BH0899*) encoding a GH11 *xylanase* from *B. halodurans* strain C-125 was overexpressed in *Kluyveromyces lactis* [12]. A cellulase-free GH11 endo-1,4-β-xylanase with a molecular mass of 24 kDa, as determined through sodium dodecyl sulfate–polyacrylamide gel electrophoresis (SDS–PAGE), was identified in *B. halodurans* PPKS-2 [13]. However, this endoxylanase exhibited maximal activity at pH 11 and 70 °C, which are slightly different from the conditions for maximal activity of the enzyme of *B. halodurans* BCRC 910501 [7].

Endoxylanases are key enzymes in the production of prebiotic XOS from biomass. XOS have varying degrees of polymerization (DP). They are oligomers composed of 2–10 xylose units through β-1,4-xylosidic linkages, and they are called xylobiose (DP2), xylotriose (DP3), xylotetraose (DP4), and so on depending on the number of xylose residues they contain. XOS can be produced from lignocellulosic biomass through alkaline extraction of xylan and subsequent enzymatic hydrolysis. Breaking the internal β-1,4-xylosidic bonds of xylan by enzymatic and/or chemical hydrolysis leads to XOS formation. Although some chemical methods are available for producing XOS [14], such as nonisothermal autohydrolysis treatment at elevated temperature, the enzymatic method is preferred in the food industry because it produces no undesirable side reactions or products [15]. Oil palm empty fruit bunch (EFB) is a type of lignocellulosic biomass produced in a large quantity during the production of palm oil; it accounts for approximately 20% of the byproducts produced during palm oil production [16]. In addition to cellulose and lignin, hemicellulose with xylan is the main component of EFB. The proportion of dry biomass in EFB varies widely, ranging from 14.62% to 33.6% (Table 1) [17–21].

**Table 1.** Chemical composition of EFB.

| Cellulose (%) | Hemicellulose (%) | Lignin (%) | Ash (%) | Extractive (%) | References |
|---|---|---|---|---|---|
| 37.26 | 14.62 | 31.64 | 6.69 | 1.34 | [17] |
| 35.8 | 19.9 * | 32.1 | - | - | [18] |
| 59.7 | 22.1 | 18.1 | - | - | [19] |
| 47.6 | 28.1 | 13.1 | - | - | [20] |
| 28.3 | 36.6 | 35.1 | - | - | [21] |

*, counted xylan only; -, not determined.

Endoxylanases from *Thermomyces lanuginosus* have been used to produce XOS from EFB [22,23]. Before enzymatic hydrolysis by an immobilized enzyme in a packed bed column reactor for XOS production, xylan was extracted from EFB in one study by using a complex process. Alkaline extraction (12% NaOH) of EFB was performed in combination with steam explosion for prolonged treatment at elevated temperature (120 °C, 2 h), followed by further hydrolysis at 25 °C for 16 h. Finally, xylan was precipitated from the

neutralized crude hydrolysate by using isopropanol [22]. In another process, a two-step chemical treatment was employed to recover the hemicellulose-rich fraction of EFB. EFB fibers were sequentially treated with peracetic acid and alkaline peroxide. Peracetic acid treatment removed approximately 50% of its lignin. The subsequent alkaline peroxide treatment recovered hemicellulose with xylan for enzymatic hydrolysis [23]. The present study explored the use of *B. halodurans* endo-1,4-β-xylanases to produce XOS from EFB, which was enzymatically reacted at high pH only after being subjected to alkali treatment.

This study determined whether XOS obtained from EFB can promote probiotic growth. Prebiotics are nondigestible food ingredients that can modulate microbiota to provide health benefits to the host. Moreover, prebiotics are considerably effective and essential for many medical applications. They do not contribute to the body's nutrition and are not absorbed but exert a profound effect on human gut flora. Prebiotics, such as XOS, can promote probiotic growth and enhance the function of probiotics to ensure a healthy balance [24]. Prebiotics could stimulate the growth of *Bifidobacterium* and *Lactobacillus* in the human intestine [25]. The main characteristics of XOS are heat resistance, acid and alkali resistance, sweet taste, and low calories. Moreover, XOS are not decomposed by bacteria in the oral cavity to produce acidic substances and cause tooth decay [26]. Recently, XOS have been recognized as novel non-digestible oligosaccharides widely used as a functional food ingredient or supplement. The global market for prebiotic XOS is growing rapidly due to their wide application in animal feed, human food additives, and medicine [27]. Given the growing global demand for prebiotics, a cost-effective method for producing XOS from various xylan-rich biomasses, including EFB, should be developed. The objective of this study was to confirm a reliable enzyme source for the efficient production of XOS from EFB. Since alkaline solutions are commonly used to dissolve xylan in EFB as a substrate, it is expected that enzymes can be applied preferentially at higher pH values. Xyn45 and Xyn23 from alkaliphilic *B. halodurans* are thus advantageous for XOS production.

## 2. Results

### 2.1. Production of Endo-1,4-β-Xylanases

*B. halodurans* BCRC 910501 was grown in Emerson medium containing both xylan and XOS, and endo-1,4-β-xylanases were synthesized and secreted into the medium. SDS–PAGE of extracellular proteins revealed bands corresponding to the endoxylanases Xyn45 and Xyn23, which have molecular masses of 45 and 23 kDa, respectively (Figure 1). Xylan can be extracted from agricultural waste, such as pineapple peel. Sodium hydroxide treatment destroyed the structure of lignocellulose and dissolved lignin and hemicellulose. Then, xylan was precipitated through soaking the alkali-soluble part in alcohol. XOS (including xylobiose and xylotriose) were obtained after enzymatic hydrolysis of xylan by endoxylanase, and both xylan and XOS could be used as substrates on which *B. halodurans* could synthesize endoxylanases. The xylanase activity per unit volume induced by pineapple peel xylan (0.5%, *w/w*) was 50.8 U/mL.

XOS containing mainly xylobiose and xylotriose induced the production of endoxylanases, and the yield of the enzyme could be improved, irrespective of the increase in the amount of xylobiose or xylotriose. To prepare the induction agent, the product generated through alkali treatment, alcohol precipitation and the enzymatic reaction of pineapple peel was filtered through a hollow fiber membrane to obtain a solution containing xylobiose (DP2) and xylotriose (DP3). The xylanase activity per unit volume increased to 135.6 U/mL after 5 days of incubation with 0.5% xylan plus 0.57% XOS (0.5% DP2 and 0.07% DP3; the xylan-DP2-DP3 ratio of 1:1:0.14 in Table 2). The specific activity was 106.1 U/mg protein. However, the production of enzymes with more XOS above this level led to reduced activity. Although the amount of protein decreased only slightly, the activity declined to 121.2 U/mL when the proportion of XOS was increased from 0.57% to 0.63% (0.5% DP2 and 0.13% DP3; the xylan-DP2-DP3 ratio of 1:1:0.26). Thus, an appropriate amount of XOS should be used as an inducer because bacteria can take up xylobiose and xylotriose and hydrolyze them

into xylose. Once the intracellular xylose concentration has increased, bacteria no longer secrete enzymes.

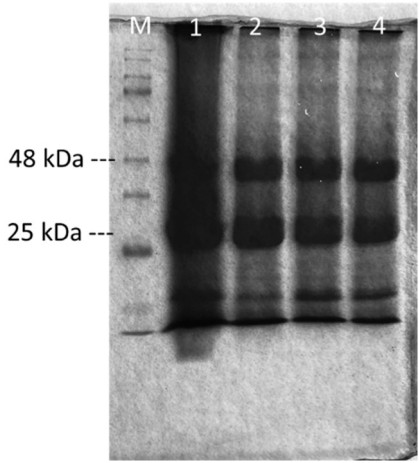

**Figure 1.** SDS–PAGE gel image of extracellular proteins produced by *B. halodurans*. Lane M is the marker (PageRuler prestained protein ladder), lane 1 is the extracellular protein induced by the combination of xylan and xylo-oligosaccharides, and lanes 2 and 4 are those induced by xylo-oligosaccharides.

**Table 2.** Concentration and activity of extracellular proteins induced by *B. halodurans* with xylan and xylo-oligosaccharides (DP2 and DP3).

| Inducer (Xylan:DP2:DP3) * | 1:0:0 | 1:0.15:0.04 | 1:0.28:0.04 | 1:1:0.04 | 1:1:0.14 | 1:1:0.26 |
|---|---|---|---|---|---|---|
| Protein concentration (mg/mL) | 1.03 ± 0.04 | 0.85 ± 0.02 | 0.88 ± 0.04 | 1.09 ± 0.25 | 1.27 ± 0.02 | 1.25 ± 0.08 |
| Activity (U/mL) | 50.8 ± 9.3 | 85.1 ± 4.5 | 89.1 ± 49.5 | 127.2 ± 54.4 | 135.6 ± 27.8 | 121.2 ± 15.8 |
| Specific activity (U/mg) | 49.6 ± 11.0 | 100.8 ± 7.5 | 102.1 ± 60.1 | 114.4 ± 41.1 | 106.1 ± 20.1 | 96.5 ± 11.6 |

* The mass ratio of xylan, DP2 and DP1 is based on 0.5% as a unit.

Recombinant endo-1,4-β-xylanase Xyn45 was produced using the bacterium harboring the plasmid-containing gene of Xyn45, *Escherichia coli* BL21(DE3)-pET29a(+)-xyn45. After induction with isopropyl β-D-1-thiogalactopyranoside (IPTG), the recombinant bacterium overexpressed the target protein Xyn45 and secreted it into the medium, suggesting that the original signal peptide from *B. halodurans* performed well in *E. coli*. When the recombinant *E. coli* was induced with 0.2 mM IPTG at 25 °C for 18 h, the harvested fermentation broth contained 0.21 mg/mL of protein and exhibited xylanase activity per unit volume of 52.1 U/mL, corresponding to specific activity of 240.2 U/mg.

### 2.2. XOS Production

XOS production was divided into two major steps. First, xylan was extracted from EFB through alkaline pretreatment. Second, the supernatant (the mixture of hemicellulose and lignin) obtained from the alkaline pretreatment was used to produce XOS through enzymatic hydrolysis. When the recombinant endo-1,4-β-xylanase Xyn45 produced by *E. coli* BL21 harboring the plasmid pET29a(+)-xyn45 was used as the catalyst, the concentration of xylobiose increased with the reaction time (Figure 2). When the xylotriose concentration increased at the beginning and then decreased in the fourth hour, it was degraded into xylobiose and xylose. After reaction for 48 h, the yield of XOS was 19.58% ± 0.52%, with a xylobiose-to-xylotriose ratio of 10:3.

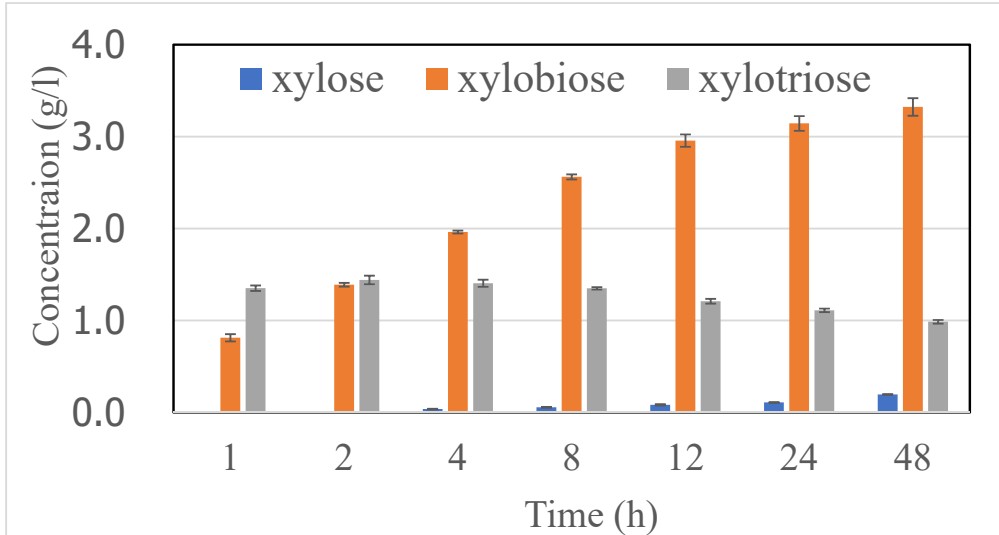

**Figure 2.** Time course of XOS production from EFB when using the recombinant endo-1,4-β-xylanase Xyn45. The enzyme dose was 0.5 U/mL. Each error bar represents the standard deviation of triplicate experiments.

When extracellular endo-1,4-β-xylanases (a mixture of Xyn45 and Xyn23) produced by wild-type *B. halodurans* were used on the basis of the same activity dose, the time course of XOS production was different. As illustrated in Figure 3, the concentrations of xylobiose and xylotriose increased with time from the beginning to the end of the reaction. A comparison of Figures 2 and 3 indicates that combined use of the endo-1,4-β-xylanases Xyn45 and Xyn23 was more efficient than use of the single endo-1,4-β-xylanase Xyn45. Because of its steric effect, Xyn45 could not cleave bonds in the internal xylan chain. However, Xyn23 could cleave bonds in the internal xylan chain. The highest yield of XOS (38.15% ± 1.30%) with a xylobiose-to-xylotriose ratio of 5:6 was obtained at 48 h through enzymatic action of the mixed endoxylanases Xyn45 and Xyn23.

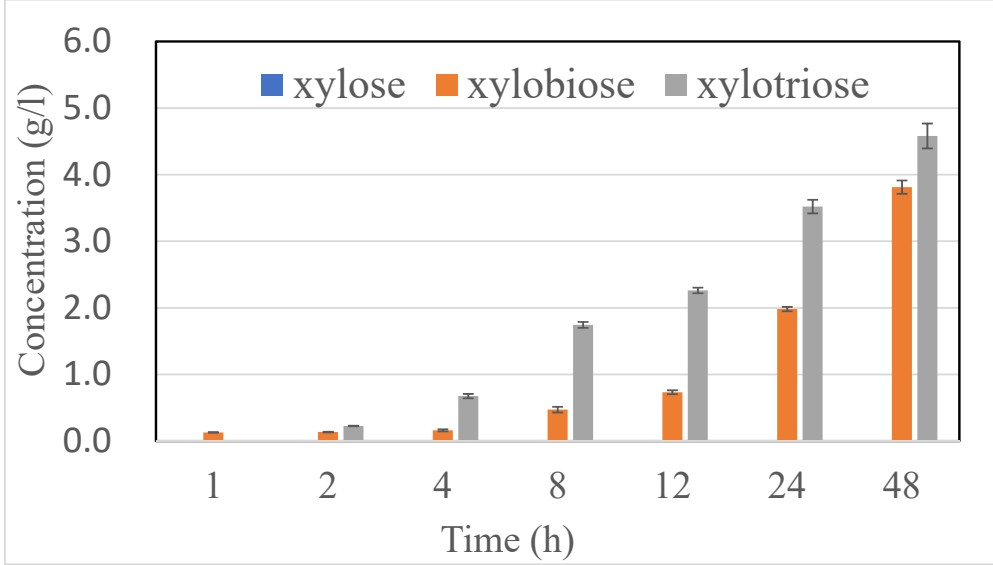

**Figure 3.** Time course of XOS production from EFB when using nonrecombinant endo-1,4-β-xylanases. The enzyme dose was 0.5 U/mL. Each error bar represents the standard deviation of triplicate experiments.

When a mixture of extracellular Xyn45 and Xyn23 produced by *B. halodurans* was used, the time course of XOS production varied depending on the dosage of the enzyme. When a low dose was used (0.5 U/mL), the concentrations of both xylobiose and xylotriose

increased with time up to 48 h (Figure 3). However, use of a high dose (2 U/mL) resulted in higher total concentrations of xylobiose and xylotriose (Figure 4). As presented in Figure 4, the concentrations of xylobiose and xylose increased with time because soluble high-DP xylo-oligomers were gradually degraded into low-DP oligomers. The xylotriose concentration increased from the beginning of the reaction to the 24th hour and then decreased because it was degraded to produce xylobiose and xylose. A comparison of Figures 3 and 4 indicates that due to the higher dosage of the enzyme, the reaction speed of the latter (Figure 4) was faster than the former (Figure 3). The yield of XOS (xylobiose and xylotriose) was highest after 24 h of reaction, reaching 46.77% ± 1.64% (w/w). The ratio of xylobiose to xylotriose was 6:5.

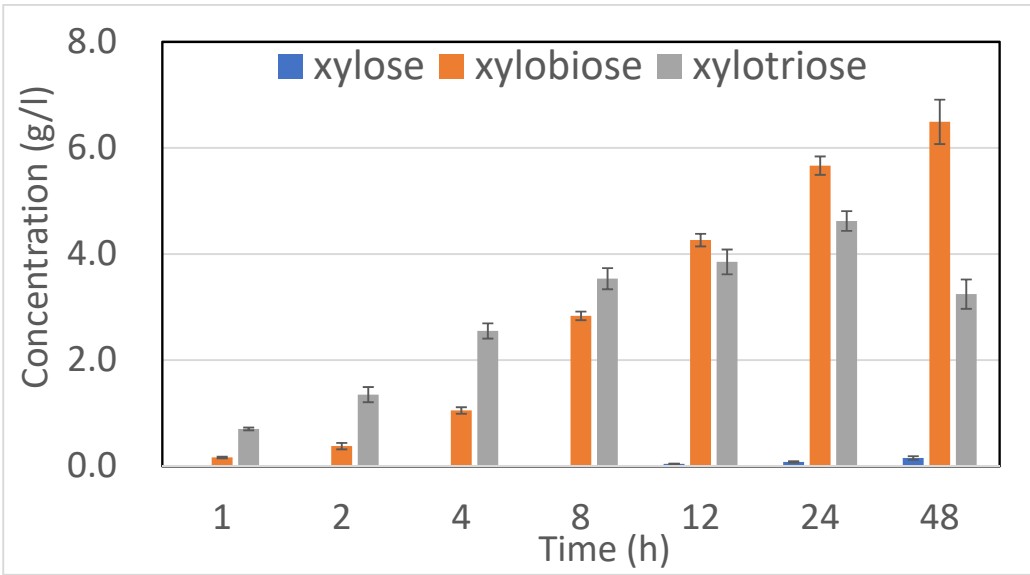

**Figure 4.** Time course of XOS production from EFB when using nonrecombinant endo-1,4-β-xylanases. The enzyme dose was 2 U/mL. Each error bar represents the standard deviation of triplicate experiments.

*2.3. Promotion of the Growth and Metabolism of Bifidobacteria and Lactobacillus*

The growth of *Bifidobacterium animalis*, *Bif. catenulatum* 14667, *Lactobacillus plantarum* 10069, and *L. acidophilus* NCFM on deMan, Rogosa, and Sharpe (MRS) medium was examined by measuring the optical density (OD) at 600 nm and the pH value. All *Bifidobacteria* and *Lactobacillus* species grew well on the MRS medium (Supplemental Table S1). *L. plantarum* 10069 had the highest growth rate on the MRS medium within 72 h, resulting in the maximum $OD_{600}$ of 1.551 ± 0.008 and the lowest pH of 3.73 ± 0.02, followed by *Bif. catenulatum 14667, L. acidophilus NCFM*, and *Bif. animalis*.

Supplemental Table S2 presents the ability of *Bif. animalis*, *Bif. catenulatum* 14667, *L. plantarum* 10069, and *L. acidophilus* NCFM to ferment glucose as a carbon source in the MRS medium. *L. plantarum* 10069 had the highest growth rate on the MRS medium containing 2% (w/v) glucose in 72 h, resulting in a maximum $OD_{600}$ of 1.543 ± 0.002 and the lowest pH of 3.41 ± 0.01, followed by *Bif. catenulatum* 14667, *L. acidophilus* NCFM, and *Bif. animalis*.

Supplemental Table S3 presents results obtained when XOS (originally 34.2 g/L with a xylobiose-to-xylotriose ratio of 6:5) from EFB instead of glucose in the MRS medium was fermented by *Bif. animalis, Bif. catenulatum* 14667, *L. plantarum* 10069, and *L. acidophilus* NCFM. *L. plantarum* 10069 had the highest growth rate on MRS medium containing 2% (w/v) XOS from EFB in 72 h, resulting in the maximum $OD_{600}$ of 1.691 ± 0.007 and the lowest pH of 4.02 ± 0.01, followed by *Bif. catenulatum* 14667, *L. acidophilus* NCFM, and *Bif. animalis*.

Figure 5 presents the comparison of $\Delta OD_{600}$ and $\Delta pH$ for *Bifidobacteria* and *Lactobacillus* growth. The four bacterial strains grew well on the MRS medium containing XOS from EFB.

Compared with glucose as a supplemented carbon source, XOS resulted in a higher OD and lower pH. Among the four bacteria, *L. plantarum* 10069 exhibited the highest growth rate and maximum pH decrease on all the media, followed by *Bif. catenulatum* 14667, *L. acidophilus* NCFM, and *Bif. animalis*. The maximum $\Delta OD_{600}$ of 1.28 and $\Delta pH$ of $-3.05$ were obtained by cultivating *L. plantarum* 10069 on the MRS medium containing XOS from EFP for 72 h. The decline in pH in the culture of the probiotic strains was attributed to the production of short-chain fatty acids. The findings indicated that the prepared XOS enhanced the growth of all the examined probiotic stains.

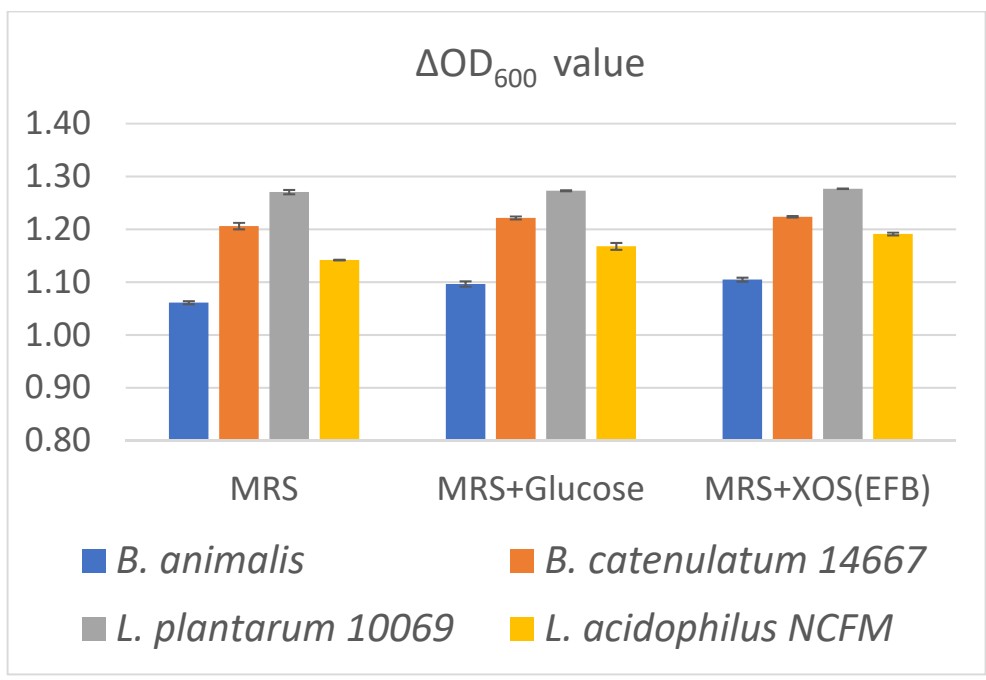

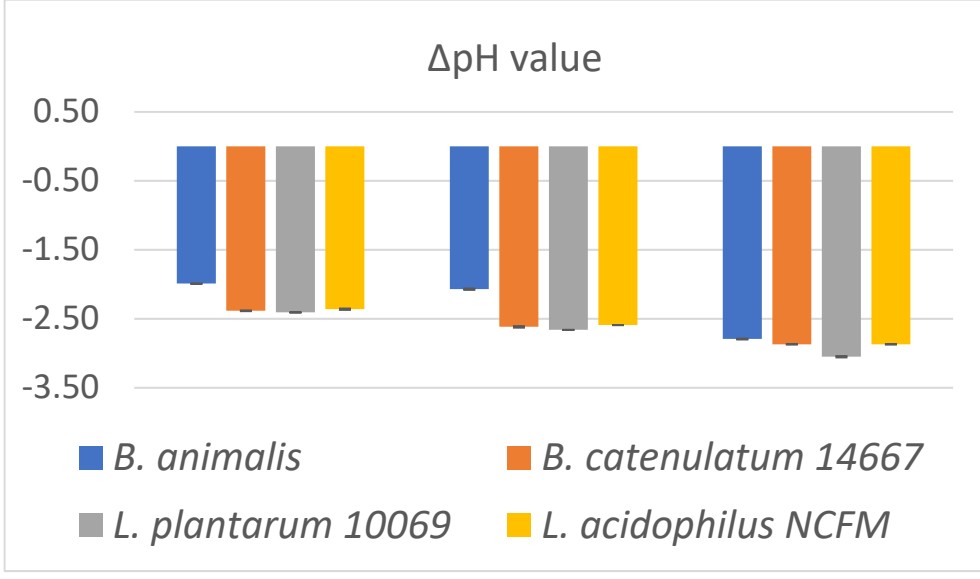

**Figure 5.** Biomass increment ($\Delta OD_{600}$) and pH decline ($\Delta pH$) of *Bifidobacteria* and *Lactobacillus* cultured in MRS medium without and with glucose or XOS for 72 h. Each error bar represents the standard deviation of triplicate experiments.

## 3. Discussion

Enzymes obtained from *Bacillus* strains are not only of academic but also industrial importance. The alkaliphilic *B. halodurans* C-125 was identified as a xylanase producer

in 1985 [28]. The results of the present study revealed that wild-type *B. halodurans* BCRC 910501 produced two extracellular endo-1,4-β-xylanases, which were active under alkaline conditions and converted lignocellulosic xylan into xylo-oligomers. In *Bacillus* and other Gram-positive bacteria, the expression of extracellular xylanases is subject to carbon catabolite repression, a regulatory mechanism by which the expression of several catabolite genes and others are repressed in the present metabolizable carbon source, such as glucose, fructose, and mannose. For example, there is a report on glucose catabolite repression on the expression of xylanase gene (*xynA*) in *B. stearothermophilus* [29]. Expression of extracellular xylanase was repressed in a medium containing glucose, suggesting that carbon catabolite repression plays a role in regulating *xynA* [30]. Thus, in this study, we used a glucose-free medium for the production of endoxylanases by *B. halodurans.* Either xylan or XOS were present in the medium as the inducer for the expression of Xyn45 and Xyn23. The end-xylanases Xyn45 and Xyn23 with molecular weights of 45 and 23 kDa, respectively, were both secreted into the extracellular space by the wild-type bacterium *B. halodurans.* Short XOS, xylobiose (DP2) and xylotriose (DP3), effectively induced the expression of endoxylanases (Table 2). Moreover, the results revealed that the induction of xylobiose was slightly better than that of xylotriose. Although studies have extensively determined the utilization of xylan by cells and carbon catabolite repression in *B. subtilis* [31–34], the regulatory mechanisms of endoxylanase genes are not yet fully understood. For example, the presence of a cis-acting catabolite responsive element in endoxylanase genes in *B. halodurans* and the interaction of this element with the global transcription regulator should be evaluated.

Although both Xyn45 and Xyn23 catalyzed hydrolysis of the linear polysaccharide β-1,4-xylan, the product patterns were slightly different. The endoxylanase in the GH10 class is usually called XynA [35]. The GH10 endoxylanase from *B. halodurans* BCRC 910501, named Xyn45, was stable over a wide pH range, including under neutral and alkaline conditions, similar to the endoxylanases from *B. halodurans* TSEV1 [36] and *B. halodurans* S7 [37]. To obtain more benefits, the endoxylanase gene should not be overexpressed in its native host cell due to the carbon catabolite repression described previously. Thus, we constructed recombinant-strain *E. coli* BL21(DE3)-pET29a(+)-xyn45 to overexpress the endoxylanase Xyn45. Higher specific activity of xylanase was achieved using recombinant bacterial cells (240.2 U/mg) compared to wild-type *B. halodurans* (114.4 U/mg, the maximum value in Table 2).

EFB is a rich source of hemicellulose, which can be extracted through alkaline pretreatment. Endoxylanases were used to generate XOS by hydrolyzing xylan chains, that is, hemicellulosic polysaccharides, in EFB. When GH10 Xyn45 [recombinant protein produced by *E. coli* BL21(DE3)-pET29a(+)-xyn45] was used alone, due to its large molecular size, it mainly acted on the ends of xylan chains. Thus, the reaction began with the generation of xylobiose and xylotriose. This is similar to the finding that XynA (a GH10 endoxylanase) in *B. halodurans* S7 mainly resulted in the production of xylobiose and xylotriose unless the reaction was performed at high temperature, such as 65 °C [38]. With an increase in time, some xylotriose was further hydrolyzed into xylose and xylobiose, and more xylobiose was produced due to the hydrolysis of terminal xylan chains. Thus, the major reaction product was xylobiose. However, at the same activity dose (0.5 U/mL) of nonrecombinant endo-1,4-β-xylanases (a mixture of Xyn45 and Xyn23 produced by *B. halodurans*), the product distribution was considerably different. In addition to the cleavage of xylan chain ends by GH10 Xyn45 to yield xylobiose and xylotriose, the action of GH11 Xyn23 generated oligomers, including xylotriose, through cleavage in the middle of xylan chains. Therefore, xylotriose was the main reaction product. As revealed in the literature, a synergistic effect could be observed in a 48 h hydrolysis experiment, and the degradation efficiency (reducing sugar generation) of the double enzyme was higher than that of the single enzyme [39]. Because of the synergistic effect of Xyn45 and Xyn23, the overall yield of XOS (the sum of xylobiose and xylotriose) after reaction for 48 h was higher than that obtained when using

a single enzyme (Xyn45). Using a mixture of enzymes nearly doubled the yield of XOS compared to using a single enzyme.

The results indicated that the reaction rate was proportional to the activity-based enzyme dose. When the dose of nonrecombinant endo-1,4-β-xylanases was increased from 0.5 to 2 U/mL, the formation of xylobiose and xylotriose was accelerated. At the 12th hour of the reaction, the concentration of xylobiose began to exceed the concentration of xylotriose, and this phenomenon was observed until the end of the reaction because xylotriose was hydrolyzed into xylose and xylotriose. At the end of the reaction, we discovered improved yield of XOS and more xylobiose than xylotriose. These experimental results indicate that by appropriately controlling the enzyme dosage and reaction time, the distribution of products can be manipulated and the yield of xylose as a byproduct can be kept very low. After 48 h of reaction, the concentrations of xylose were respectively 0.195 ± 0.004 g/L, 0 g/L, and 0.156 ± 0.034 g/L, corresponding to the data in Figures 2–4.

XOS are prebiotics that confer a health benefit on hosts by modulating microbiota. XOS can reduce intestinal pathogens, blood cholesterol, triglycerides, and glucose and can enhance antioxidant activity. Enhancement of the growth and metabolic activity of prebiotic bacteria can be quantified by an increase in OD and a decrease in pH. The decline in pH can be due to the production of short-chain fatty acids, the main metabolites produced by probiotics. The four probiotic bacteria investigated in this study—*Bif. animalis*, *Bif. catenulatum* 14667, *L. plantarum* 10069, and *L. acidophilus* NCFM—grew well on MRS medium containing XOS from EFB. Among them, *L. plantarum* 10069 exhibited the highest growth on the XOS-containing medium. Similarly, XOS were tested as a prebiotic for two probiotic bacteria, and XOS more efficiently enhanced the growth of *L. plantarum* WU-P19 than that of *Bif. bifidum* TISTR 2129 [23]. The ability of *Lactobacillus* strains to compete with other gut bacteria can be determined by examining their XOS utilization pattern [40]. In summary, XOS caused an increase in cell mass (OD) and a decrease in pH, suggesting that XOS enhanced the growth of all the studied probiotic strains. XOS from other types of lignocellulose waste have been demonstrated to exert similar prebiotic effects [41–43].

## 4. Materials and Methods

### 4.1. Preparation of Nonrecombinant Endoxylanases

Two types of enzymes were prepared in this study: (1) nonrecombinant endo-1,4-β-xylanases (a mixture of Xyn45 and Xyn23) produced by wild-type *B. halodurans* and (2) recombinant endo-1,4-β-xylanase Xyn45.

To produce the mixture of nonrecombinant endoxylanases, *B. halodurans* BCRC 910501 was grown on xylan-containing medium to induce the production of extracellular proteins. An inoculum was prepared by growing the bacterium in 5 mL of basal salt medium (composed of 1 g/L $K_2HPO_4$, 5 g/L NaCl, 0.2 g/L $MgSO_4 \cdot 7H_2O$, 5 g/L peptone, 2 g/L yeast extract, and 10 g/L glucose) at 37 °C for 16 h. Then, 5 mL of the precultured bacterial solution was poured into 100 mL of glucose-free Emerson medium (composed of 1 g/L $K_2HPO_4$, 0.2 g/L $MgSO_4 \cdot 7H_2O$, 5 g/L peptone, and 5.5 g/L yeast extract; pH 10) supplemented with xylan or a mixture of xylan and XOS from pineapple peel as the sole carbon source. The culture was grown at 37 °C with shaking at 175 rpm for 5 days. Then, the fermentation broth was centrifuged, and the obtained supernatant contained extracellular proteins secreted by bacterial cells into the culture medium.

Xylan was prepared from pineapple peel through alkaline pretreatment followed by ethanol precipitation. Briefly, pineapple peel waste was soaked in 4% sodium hydroxide solution at a solid-to-liquid ratio of 1:15 (*w/v*) in an incubator; the waste was incubated at 30 °C with shaking 100 rpm for 24 h. Subsequently, after centrifugation at $2560 \times g$ to separate the precipitate out of the solution, the supernatant was adjusted to pH 7 by using concentrated hydrochloric acid. Three volumes of 95% ethanol were added to the supernatant, and the mixture was placed in a refrigerator at 4 °C. After 24 h, the sample was centrifuged to separate the precipitate (hemicellulose) from the solution. The precipitate

(xylan) was dried in an oven and then used to induce the production of endoxylanases by *B. halodurans*.

Combinations of xylan and XOS containing certain amounts of xylobiose and xylotriose were also used for the induction of endoxylanases. To produce xylobiose and xylotriose, xylan obtained from pineapple peel was hydrolyzed using the recombinant endoxylanase Xyn45 (prepared in accordance with the protocol described in the next section). Hemicellulose (xylan) from pineapple peel was mixed with 100 mM Tris HCl buffer (pH 8.0) to obtain 2% (*w/w*) hemicellulose solution. Then, 2 U/mL Xyn45 was added, and the reaction was conducted at 50 °C in a water bath with shaking at 50 rpm for 24 h. Subsequently, the reaction mixture was placed in a water bath at 100 °C for 5 min to inactivate the enzyme and then centrifuged at 2560× *g* at 25 °C for 20 min. The supernatant was filtered using a 0.22-μm filter to yield XOS solution. After it was decolorized using the anion exchange resin HYDROLUX S5458, the XOS solution was ultrafiltered through a 1-kD hollow fiber membrane to a tenth of the original volume of the filtrate. High-DP oligomers and unhydrolyzed xylan were removed from the XOS solution. Finally, the filtrate was concentrated in a vacuum concentrator, and the concentration of each oligosaccharide in it was measured through high-performance liquid chromatography (HPLC) performed on a Rezex RSO-Oligosaccharide Ag+ 4% column (Phenomenex, Torrance, CA, USA).

### 4.2. Preparation of Recombinant Endoxylanase Xyn45

Recombinant *B. halodurans* endo-1,4-β-xylanase was prepared from *E. coli* BL21(DE3)-pET29a(+)-xyn45. To construct the recombinant plasmid, the gene encoding Xyn45 in *B. halodurans* was amplified through polymerase chain reaction (PCR) by using the forward primer 5′-GGA ATT CCA TAT GAT TAC ACT TTT TAG AAA GCC TTT TGT TGC TGG G-3′ and the reverse primer 5′-CCG CTC GAG CTA ATC AAT AAT TCT CCA GTA AGC AGG TTT CAC TCG-3′. The PCR-amplified product was purified and double-digested using the restriction endonucleases NdeI and XhoI. Furthermore, the vector plasmid pET-29a(+) was purified and double digested using the same restriction endonucleases. After ligation of the PCR-amplified product and vector plasmid, the resulting recombinant plasmid was transformed into *E. coli* DH5a. This recombinant plasmid was named pET-29a(+)-xyn45. Confirmed by DNA sequencing, it was transformed into *E. coli* BL21 (DE3) to yield a recombinant strain called *E. coli* BL21(DE3)-pET29a(+)-xyn45.

To produce the endoxylanase Xyn45, 5 mL of the recombinant bacterial preculture was inoculated into 50 mL of PM medium containing 0.05 mg/mL kanamycin. The bacterial culture was grown at 37 °C with shaking at 175 rpm. When absorbance at 600 nm reached 0.8–1.0, IPTG was added to induce recombinant protein expression for 18 h at 25 °C. After this induction, the bacterial solution was centrifuged at (4260× *g*) at 4 °C for 20 min, and the obtained supernatant was the extracellular protein solution.

### 4.3. Analysis of Enzyme Activity and Protein Assay

The activity of endo-1,4-β-xylanase was examined on the basis of the production of reducing sugars from xylan by using the 3,5-dinitrosalicylic acid (DNS) method. In two test tubes labeled A1 and A2, 50 μL of the enzyme was mixed with 450 μL of 1% birch xylan (in 100 mM Tris-HCl buffer, pH 8). The A2 test tube was placed in a water bath at 60 °C for 10 min, whereas simultaneously, the A1 test tube was placed in an ice bath for 10 min. Then, 1 mL of DNS reagent was added to A1 and A2. The A1 and A2 test tubes were placed in a water bath at 100 °C for 10 min to inactivate the enzyme. A spectrophotometer was used to measure the absorbance at 540 nm. The xylose calibration curve (y = 1.095x − 0.026) was used to convert the OD at 540 nm into the concentration of reducing sugars. Enzyme activity (U, in terms of $\mu$ mole/min) was calculated using the formula U = (A2 − A1)/Δt.

Extracellular proteins were analyzed through SDS–PAGE by using 12.5% acrylamide gel. The resolved protein bands were visualized through Coomassie brilliant blue staining. The amount of protein was quantified using the Bradford assay based on the use of Coomassie Brilliant Blue G-250, and bovine serum albumin was used as the standard protein.

### 4.4. XOS Production from EFB through Enzymatic Reaction

EFB was provided by Southern Palm (1978), Surat Thani, Thailand. The method developed by National Renewable Energy Laboratory (NREL) [44] was used to determine the composition of lignocellulose in raw materials. EFB was composed of 22.4% cellulose and 22.0% hemicellulose based on dry mass.

First, the raw EFB was washed with tap water to remove soil and other impurities from its surface. Then, it was dried in an oven at 60 °C for 24 h and pulverized to obtain powder with a particle size of 40–60 mesh. The EFB powder was soaked in 15% sodium hydroxide solution at a solid-to-liquid ratio of 1:10 (*w/v*) in an incubator and then incubated at 45 °C with shaking at 100 rpm overnight. Subsequently, the mixture was centrifuged ($2400 \times g$ at 25 °C for 20 min) to separate the precipitate (cellulose) out of the solution. The supernatant (hemicellulose and lignin) was adjusted to pH 8 by using concentrated hydrochloric acid.

The solution was then treated with either recombinant *B. halodurans* endo-1,4-β-xylanase (from *E. coli* BL21-pET 29a-Xyn45) or extracellular endo-1,4-β-xylanases (from *B. halodurans* BCRC 910501) for the enzymatic hydrolysis of EFB-derived xylan in the solution. Enzyme preparations at different concentrations (0.5 U and 2 U for 1 mL of the sample) were used. The enzyme reaction was conducted in a water bath at 50 °C with shaking at 50 rpm for 48 h. Samples were removed at fixed intervals (1, 2, 4, 8, 12, 24, and 48 h), and the enzymatic reaction was inactivated by boiling the sample in a water bath at 100 °C for 10 min. After boiling, the sample was centrifuged ($2400 \times g$ at 25 °C for 20 min) to separate out the precipitate (lignin) and keep the supernatant as the product (XOS). Finally, the supernatant was passed through a 0.22 μm filter by performing vacuum filtration. Then, the XOS concentration was determined through HPLC.

### 4.5. Fermentation of Probiotics on XOS

Ultrafiltration was performed using a hollow fiber membrane with a 1-kDa molecular weight cutoff to remove the enzyme and high-molecular-weight polysaccharides (DP5 and larger than DP5) from the XOS products after enzymatic hydrolysis. The permeate was concentrated by boiling to remove water.

Four bacterial strains were used in this study: *Bif. animalis*, *Bif. catenulatum* 14667, *L. plantarum* 10069, and *L. acidophilus* NCFM. The bacteria were cultured in MRS broth containing 34.15 g/L of Lactobacilli MRS broth, 0.5 g/L of cysteine monohydrochloride monohydrate, and 1 mL/L Tween #80. For each inoculum, 6 mL of MRS broth and 1 mL of mineral oil were added to a glass test tube. After being capped, the glass test tube was kept sterile at 121 °C for 20 min and then naturally cooled to room temperature. In a laboratory fume hood, 1 mL of lactic acid bacterial liquid (containing *Bifidobacterium or Lactobacillus*) was inoculated and stored at −80 °C in a glass test tube. Finally, the glass test tube was incubated anaerobically at 37 °C for 24 h.

Fermentation experiments were performed using different media (MRS broth, MRS broth containing 2% *w/v* glucose, and MRS broth containing 2% *w/v* of XOS from EFB). For each fermentation, 30 mL of MRS broth and 2.5 mL of mineral oil were added to a 50-mL sharp-bottomed centrifuge tube. After being capped, the sharp-bottom centrifuge tube was sterilized at 121 °C for 20 min and then naturally cooled to room temperature. The laboratory fume hood was used for inoculating 1 mL of the inoculum into the sharp-bottom centrifuge tube. Finally, the glass test tube was incubated anaerobically at 37 °C for 72 h. Bacterial growth was analyzed by measuring absorbance at 600 nm and pH at 0, 24, 48, and 72 h.

## 5. Conclusions

Both the recombinant and nonrecombinant endo-1,4-β-xylanases of alkaliphilic *B. halodurans* BCRC 910501 effectively converted the xylan within EFB into prebiotic XOS. The synergistic effect of the two endoxylanases Xyn45 and Xyn23—belonging to the GH10 and GH11 families of enzymes, respectively—enabled faster cleavage of xylan into small-molecule XOS. The yield of XOS was higher when a mixture of Xyn45 and Xyn23 was used

compared with when a single enzyme, Xyn45, was used. The recombinant endoxylanase Xyn45 could be produced by recombinant *E. coli* BL21-pET 29a(+)-xyn45, while a mixture of nonrecombinant endoxylanases Xyn45 and Xyn23 could be obtained by culturing *B. halophilus* in a medium containing xylan and XOS. Xylobiose in XOS was more effective than xylotriose in inducing *B. halophilus* to secrete endoxylanases. The EFB-derived XOS promoted the growth and metabolism of probiotic strains, *Bifidobacteria* and *Lactobacilli*. The feasibility of the production of these two endoxylanases and their uses indicate their potential for industrial application.

**Supplementary Materials:** The following supporting information can be downloaded at: https://www.mdpi.com/article/10.3390/catal13010039/s1, Table S1: Cultivation of *Bifidobacteria* and *Lactobacillus* on MRS medium; Table S2: Cultivation of *Bifidobacteria* and *Lactobacillus* on MRS medium containing 2% (*w/v*) glucose; Table S3: Cultivation of *Bifidobacteria* and *Lactobacillus* on MRS medium containing 2% (*w/v*) XOS from EFB.

**Author Contributions:** Conceptualization, W.-C.L., N.L. and C.-Y.C.; data curation, C.T. and Y.-C.H.; methodology, C.T., Y.-C.H. and J.-M.S.; software, C.T. and Y.-C.H.; visualization, C.T.; writing—original draft, W.-C.L.; writing—review and editing, N.L. and C.-Y.C.; supervision, W.-C.L. All authors have read and agreed to the published version of the manuscript.

**Funding:** We thank the Taiwan Ministry of Science and Technology (Grant number: MOST 109-2221-E-194-014) for supporting this study.

**Data Availability Statement:** Data reported in this study were collected and stored by C.T and Y.-C.H.

**Acknowledgments:** We would like to express our gratitude to Kow-Jen Duan at Tatung University, Taiwan for providing the probiotic strains.

**Conflicts of Interest:** The authors declare no conflict of interest.

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
