# Peer review of "Application of Endoxylanases of Bacillus halodurans for Producing Xylooligosaccharides from Empty Fruit Bunch"

_catalysts, doi:10.3390/catal13010039_

Round 1

Reviewer 1 Report

This paper deals with the xylo-oligosaccharides using endoxylanases.  This paper contains new results, so that the paper can be published after the following points are revised:

     The merit of this paper is not clear.  The authors should be shown the importance of xylo-oligosaccharides.  Additionally, the advantage points of this paper is not shown.  The authors should discuss the point of the present method compared with the others.

Especially, the authors should show why enzymes of Xyn45 and Xyn23 were employed.

     The authors wrote "the synergistic effect" at line 292 would cause the enhancement of production yield.  The authors should discuss the detail of the effect.

     Some technical expression would be different.  For example, "enzyme preparation" in Abstract would be "prepared enzyme" or "purified enzyme".  The authors should check the throughout the manuscript.

     There is no Supplemental Table at line 213 and 219.

Author Response

Response to Comments of reviewer 1:

This paper deals with the xylo-oligosaccharides using endoxylanases.  This paper contains new results, so that the paper can be published after the following points are revised:

—     The merit of this paper is not clear.  The authors should be shown the importance of xylo-oligosaccharides.  Additionally, the advantage points of this paper is not shown.  The authors should discuss the point of the present method compared with the others.

Especially, the authors should show why enzymes of Xyn45 and Xyn23 were employed.

Ans: At your suggestion, in the Introduction of the revised manuscript we added some text explaining the importance of xylo-oligosaccharides (with a new reference 27) and the advantage of using xyn45 and xyn23.

—     The authors wrote "the synergistic effect" at line 292 would cause the enhancement of production yield.  The authors should discuss the detail of the effect.

Ans: In the revised manuscript, we have added a sentence and a reference paper (new ref. 38) to additionally address the synergistic effect.

—     Some technical expression would be different.  For example, "enzyme preparation" in Abstract would be "prepared enzyme" or "purified enzyme".  The authors should check the throughout the manuscript.

Ans: The term "enzyme preparation" in Abstract has been rephrased as "prepared enzyme" at your suggestion.

—     There is no Supplemental Table at line 213 and 219.

Ans: We have added the Supplemental Tables to the revised manuscript at the end. We apologize for this omission. 

Reviewer 2 Report

Valorizing the wastes in producing new compounds, especially enzymes that can hydrolyze cellulose. It is an inserting topic.

In lines 16 and 17, the authors give two codes, XYn45 and Xy23; you indicate xyn 45 twice, where the xyn23 is. Clear each code beside its enzymes to be easy for the reader

Check the abstract for linguistic mistakes and throughout the manuscript; more sentences are difficult

add more results in the abstract

Clear the aim and recommendation in the abstract

In the introduction, xyn45 was introduced, whereas the syn 23

Add an electropherogram image of two enzymes

Clear the objectives of your study at the end of the introduction

Enhance Figure 1

Line 143, use the abbreviation only you indicated previously

Redesign the heads of Table 2 as a ratio between inducers, for example, the second column 7:1:1, then indicate the inducers in the table footnote

Figure 3 convert into a Table to show the values of xylose

Also, figure 4

Figure 5 scientific name must be italic in the figure legend

Line 383 delete the bold

Provide the origin and city of all devices in manuscript

Enhance conclusion

Check the outputs of all references

Author Response

Response to Comments of reviewer 2:

Valorizing the wastes in producing new compounds, especially enzymes that can hydrolyze cellulose. It is an inserting topic.

In lines 16 and 17, the authors give two codes, XYn45 and Xy23; you indicate xyn 45 twice, where the xyn23 is. Clear each code beside its enzymes to be easy for the reader

Ans: Xyn45 and Xy23 are GH10 xylanase and GH11 xylanase, respectively. We have indicated these in the abstract.

Check the abstract for linguistic mistakes and throughout the manuscript; more sentences are difficult

add more results in the abstract

Clear the aim and recommendation in the abstract

Ans: We are unable to add more content due to the word limit on the abstract. However, we did our best to revise it based on your suggestions.

In the introduction, xyn45 was introduced, whereas the syn 23

Add an electropherogram image of two enzymes

Clear the objectives of your study at the end of the introduction

Ans: Both Xyn45 and Xy23 have been well introduced in the second paragraph of Introduction. We have included images of SDS-PAGE gel electrophoresis of proteins synthesized by B. halodurans as shown in Figure 1. In Figure 1, two enzymes Xyn45 and Xy23 can be seen. At your suggestion, we have added more to include the objectives of this study at the end of the introduction.

Enhance Figure 1

Ans: We are sorry that lane 1 in Figure 1 is blurred together due to too much protein loaded on the gel, but in other lanes 2-4, we can clearly see the bands of these two enzyme proteins, Xyn45 and Xy23.

Line 143, use the abbreviation only you indicated previously

Ans: The abbreviation of degree of polymerization (DP) has indicated in the Introduction.

Redesign the heads of Table 2 as a ratio between inducers, for example, the second column 7:1:1, then indicate the inducers in the table footnote

Ans: The heads of Table 2 have been changed to a mass ratio of xylan, DP2 and DP1 at your suggestion.

Figure 3 convert into a Table to show the values of xylose

Also, figure 4

Figure 5 scientific name must be italic in the figure legend

Ans: At the end of 4th paragraph of Discussion, we have reported in the revised manuscript the values of xylose concentration read from Figure 3 and Figure 4. The scientific name of probiotics has been italic in Figure 5 at your suggestion.

Line 383 delete the bold

Provide the origin and city of all devices in manuscript

Enhance conclusion

Check the outputs of all references

Ans: We have made changes on manuscript in response to these comments.

Reviewer 3 Report

The authors report a work on the production of xylooligosaccharides (XOS) by endoxylanases using oil palm empty fruit bundles (EFB) as the initial feedstock. The catalytic efficiency of the recombinant endonuclease and the non-recombinant endonuclease were compared in the production of XOS. They demonstrated that when the two endonucleases were combined, there was a synergistic effect on the hydrolysis of EFB, producing more XOS. In addition, according to their experimental evidence, XOS proved to be a prebiotic and promoted the growth of probiotics. This study is full of significance and has important implications for the reuse of natural resources. There are some suggestions for minor revisions to the manuscript before acceptance.

1.      It is not an ecologically and economically friendly way to obtain the EFB-derived xylan by using strong bases and strong acid chemicals. Therefore, it would increase the value of this work by exploring more environmentally and economically friendly pathways, such as producing XOS directly from EFB as the feedstock.

2.      The scheme of xylan to XOS can be drawn to demonstrate the process of enzymatic hydrolysis.

3.      Page 3, line 127, double-check the following statement to make sure it is expressed correctly, i.e., “... enzymatic hydrolysis of endoxylanase by xylan, ...”

4.      In table 1, add the annotations of symbol “*” and “-”; In figure 1, Where are the annotations of lane 4? In Figure 2 to figure 5, give the number of data repetitions for the error bars.

Author Response

Response to Comments of reviewer 3:

The authors report a work on the production of xylooligosaccharides (XOS) by endoxylanases using oil palm empty fruit bundles (EFB) as the initial feedstock. The catalytic efficiency of the recombinant endonuclease and the non-recombinant endonuclease were compared in the production of XOS. They demonstrated that when the two endonucleases were combined, there was a synergistic effect on the hydrolysis of EFB, producing more XOS. In addition, according to their experimental evidence, XOS proved to be a prebiotic and promoted the growth of probiotics. This study is full of significance and has important implications for the reuse of natural resources. There are some suggestions for minor revisions to the manuscript before acceptance.

  1. It is not an ecologically and economically friendly way to obtain the EFB-derived xylan by using strong bases and strong acid chemicals. Therefore, it would increase the value of this work by exploring more environmentally and economically friendly pathways, such as producing XOS directly from EFB as the feedstock.

Ans: We totally agree with you that using strong acids or bases is unfriendly to the environment. However, as indicated in reference 27 (new), alkali extraction overall has the advantages of a good separation effect and higher yield for XOS production. Proper recovery of alkali during the process should be a way to reduce the impact on the environment.

  1. The scheme of xylan to XOS can be drawn to demonstrate the process of enzymatic hydrolysis.

Ans: Both xylan and XOS are molecules linked by xylose via beta-1,4 linkage. The only difference is the number of linked xylosyl groups. Breaking simply the internal β-1,4-D-xylosidic bonds of xylan yields XOS. So we don't show the scheme of xylan to XOS, but just describe it with the cited references in the third paragraph of the Introduction.

  1. Page 3, line 127, double-check the following statement to make sure it is expressed correctly, i.e., “... enzymatic hydrolysis of endoxylanase by xylan, ...”

Ans: This syntax error has been corrected. Thanks for your correction.

  1. In table 1, add the annotations of symbol “*” and “-”; In figure 1, Where are the annotations of lane 4? In Figure 2 to figure 5, give the number of data repetitions for the error bars.

Ans: At your suggestion, the annotations of symbol have been given in Table 1 and the caption of Figure 1 has been corrected. The number of data repetitions for the error bars has been added to the captions of Figures 2 to 5.

Reviewer 4 Report

It is an interesting work. However explanations for results can be improved.

Author Response

Response to Comments of reviewer 4:

It is an interesting work. However explanations for results can be improved.

Ans: Thank you for your comment. We have tried our best to revise the manuscript.

Round 2

Reviewer 1 Report

The paper was revised along with the referees' comments, so that the paper can be published.

Reviewer 2 Report

The manuscript now can be accepted